# Knowledge and Protective Behaviors of Teachers on Skin Cancer: A Cross-Sectional Survey Study from Turkey

**DOI:** 10.3390/children10020291

**Published:** 2023-02-02

**Authors:** Celal Kus, Mine Mujde Kus, Hamit Sirri Keten, Huseyin Ucer, Numan Guvenc, Fatih Kus, Hasan Cagri Yildirim, Hatice Tuba Akbayram

**Affiliations:** 1Department of Family Medicine, Medical Faculty, Kahramanmaras Sutcu Imam University, Kahramanmaras 46050, Turkey; 2Department of Dermatology and Venereology, Medical Faculty, Kahramanmaras Sutcu Imam University, Kahramanmaras 46050, Turkey; 3Department of Family Medicine, Medical Faculty, Gaziantep University, Gaziantep 27050, Turkey; 4Department of Family Medicine, Pazarcik Family Health Center (No. 1), Kahramanmaras 46050, Turkey; 5Department of Family Medicine, Pazarcik Family Health Center (No. 2), Kahramanmaras 46050, Turkey; 6Department of Medical Oncology, Hacettepe University Medical School, Ankara 06230, Turkey

**Keywords:** skin cancer, primary prevention, sun safety, teacher, education

## Abstract

Background: People socialize and receive education and training for the first time outside the home at school, where their teachers act as role models. Teachers play a crucial role in instilling sun-protection habits in children. Avoiding the sun between 10 a.m. and 4 p.m., staying in the shade, wearing sun-protective clothing, wearing sunglasses, wearing hats, using sunscreen products, and using an umbrella are some of the sun-protection methods described in the literature. This study sought to assess teachers’ skin cancer knowledge and attitudes (SC). Methods: In this cross-sectional study conducted between 21 September 2020, and 21 October 2020, 647 teachers from 30 schools in Kahramanmaras were included with their consent. The number of teachers employed at Kahramanmaras was 1863. Accordingly, the sample was found to be 641 with a 5% margin of error and a 99% confidence interval. Schools were selected by a simple random method. Teachers’ knowledge and behaviors were assessed using a 25-point questionnaire designed by the literature data to gauge the level of SC knowledge. Results: Of the 647 teachers included in this study, 230 (35.5%) were male, and 417 (64.5%) were female. The mean age of the participants was 38.44 ± 8.79 (min = 22, max = 65) years. The knowledge level of the teachers about SC was 13.54 ± 4.48 (min = 0, max = 23). The internet (75.9%) was the most preferred source of information. SC knowledge level was significantly higher in those with SC in their families and birthmarks on their bodies. (*p* < 0.001, *p* = 0.042, respectively). The rate of precaution taken to protect from the sun was higher in those with higher knowledge (*p* = 0.032). Women, primary school teachers, those with skin type 1, those with multiple nevi, and those with a high level of SC knowledge used statistically significantly more sunscreen (*p* = 0.001, *p* = 0.003, *p* < 0.001, *p* = 0.037, *p* = 0.002, respectively). Conclusions: It was found that the knowledge level of teachers about skin cancer and sun-protective behaviors was moderate. Correct behaviors increased as knowledge about SC grew. Information and recommendations made on the Internet should be made by experts. Additionally, health policymakers should implement projects aimed at improving teachers’ knowledge and behaviors and, through them, teaching students about SC; as such projects would significantly contribute to both public health and health economics.

## 1. Introduction

In many countries, SC (skin cancer) is one of the most frequently diagnosed cancers. SC is classified into two main categories: melanoma and non-melanoma SC. Non-melanoma SC refers to all types of non-melanoma cancers occurring in the skin, primarily basal cell carcinoma and squamous cell carcinoma [1].

New SC cases and deaths are increasing day by day [1,2,3]. In 2018, there were 287,723 new cases of melanoma and 60,712 deaths related to this cancer globally. Additionally, in 2018, there were 1,042,046 new cases of non-melanoma SC and 65,155 deaths due to these cancers worldwide. In Europe, there were 318,345 new cases of SC and 13,769 deaths, with a 5-year prevalence of 1,628,092 people. In Turkey, the 5-year prevalence of skin cancer is 4809, and there were 1622 new cases and 669 deaths related to SC in 2018 [2,3].

Cancer incidence and cancer-related deaths are affected by risk factor exposure, screening tests, and treatment modalities [2]. Approximately 5 million people in the United States receive treatment for SC each year. SC causes severe economic, social, and health problems [2]. SC can be significantly prevented by protection from risk factors. The most effective way to reduce SC risk is to protect against the sun, a significant risk factor. Avoiding the sun between 10 a.m. and 4 p.m., staying in the shade, wearing sun-protective clothing, wearing sunglasses, wearing a hat, using sunscreen products, and using an umbrella are some of the most effective sun-protective methods [2]. The ultraviolet (UV) index measures the intensity of UV radiation (UVR), a measurement that has been used for about 20 years. UV index exposure levels are associated with the risk of sunburn and SC [4,5]. When the UV index is 3 or above, there is a need to use sunscreen as a precaution. In 2020, our region’s average UV index was 7–8 in summer, 5 in spring, and 2 in winter. Although there is no need for strict sun-protection measures in winter in our region, the UV index is above 3 at noon on sunny days in winter. As a result, those with SC risk factors and photosensitivity should use sunscreen during these days and hours [6,7]. Because children’s skin is thinner and more susceptible to sunburn, protecting children from the sun’s harmful UVR exposure is one of the most effective ways to prevent future skin cancer developmen [8]. Children’s education and training hours coincide with the hours of highest UVR [7]. Teachers are role models for students and their influence on individuals’ education increases the importance of their knowledge about SC and their protective behaviors about SC [9,10].

Due to its proximity to the equator and the light skin color of the population, Australia’s incidence of SC is relatively high [11,12]. For this reason, many studies and public health campaigns have been carried out to prevent SC in Australia. In Turkey, although not as comprehensively as in Australia, information is given on this subject. Every year in Turkey, May has been accepted as the month of skin cancer awareness. During this month, posters are displayed and seminars are held in hospitals, though usually only for hospital and provincial health department employees. Additionally, informational brochures on sun protection, self-examination of nevi, and skin cancer, organized by the Turkish Dermatology Association, can be accessed on the association’s website [13].

Many studies have evaluated students’ knowledge and behaviors on this topic. These have led to increased interventions to prevent skin cancer in some countries [14,15,16,17,18,19,20]. However, few studies evaluate the knowledge levels and behaviors of teachers and preservice teachers who undertake the education and training of students and are role models for them regarding SC and sun exposure, which is the most important preventable risk factor [11,21,22,23,24,25].

Studies with teacher candidates show that providing this SC training before they begin the profession improves students’ sun safety [11]. Turkey is one of the countries where public health policies for sun safety, which is the most important preventable risk factor for skin cancer, have not yet been established. In the literature, studies evaluating the knowledge and behaviors of teachers and preservice teachers about skin cancer and sun safety were mostly conducted in countries where these public health policies were developed [11,21,22,23,24,25]. We came across two studies conducted for teachers on this subject in our country [23,24,25,26]. Studies to be made on this subject will guide policies to improve solar safety for countries such as ours, where policies have not yet been developed in this regard, and where year-round UV index averages are high. This study aimed to evaluate the knowledge level of teachers about SC and their sun-protective behaviors.

## 2. Materials and Methods

This cross-sectional study was carried out between 21 September 2020, and 21 October 2020, in Kahramanmaras in the Mediterranean region of Turkey. This region has an average UV index of 4.75 throughout the year and an average sunshine duration of 8.7–10 h between June and September. There were 18,603 teachers employed at Kahramanmaras [27]. Accordingly, the sample was found to be 641, with a 5% margin of error and a 99% confidence interval. Six hundred and forty-seven teachers participated in this study. Schools were selected by a simple random method. Teachers in 30 schools in Kahramanmaras took part in this study. Based on the literature review on skin cancer, the researchers created a 40-question, three-part questionnaire. The questionnaire was applied to 25 people and rearranged according to the feedback. The questionnaire was then emailed to the teachers via Google Forms. Surveys were implemented with Google Forms as it would be difficult to conduct in-person surveys for geographical reasons. Before this study, teachers were informed via e-mail and telephone applications. A consent form and questionnaire, prepared via Google Forms, were sent to those who wanted to take part in this study. Teachers who gave consent and completed the questionnaire were included in this study. Teachers who gave incomplete answers to the skin cancer knowledge questions were excluded from this study.

The first part of the questionnaire included questions about the sociodemographic (age, gender, marital status, and if they had children) and occupational characteristics of the teachers, and the second part of the questions gathered information on the teachers’ skin characteristics (skin types, plenty of nevi, and birthmarks on the skin), sun-protection methods on sunny days, number of sunburns, and family history (SC). “Plenty nevi” was defined as the presence of more than 50 nevi. Skin types were shown according to the participant’s responses with the Fitzpatrick classification, which Thomas B. Fitzpatrick developed to predict the response of different skin types to ultraviolet (UV) light [28].

In the third part of the questionnaire, questions revealed teachers’ general knowledge about SC, its risks, symptoms, and treatment. The reliability of knowledge questions on SC was assessed using Cronbach’s internal consistency coefficient. The Cronbach alpha value was 0.799.

### Statistical Analysis

All statistical analyses were performed using SPSS version 22.0 software. In the analysis of the data, frequency, mean, and standard deviation values were shown. The Kolmogorov–Smirnov test was used to determine whether the variables were compatible with a normal distribution. Student’s t- and Mann–Whitney U tests were used to reveal the difference between the two groups. A one-way variance analysis was applied to compare three or more groups. *p*-value < 0.05 was defined to be significant.

Twenty-five questions based on literature data were asked to assess the teachers’ SC knowledge level. The correct answer to each knowledge question was evaluated as one point, and the total knowledge level was defined as 25 points (for 25 questions). In the questions prepared on the triple Likert scale, the statements “I agree” for correct statements about SC and “I disagree” for incorrect statements received points. The “I have no idea” statement was not scored.

## 3. Results

Of the 647 teachers included in this study, 230 (35.5%) were male and 417 (64.5%) were female. The mean age of the teachers was 38.44 ± 8.79 (min = 22, max = 65), and the average duration of professional service was 14.54 ± 8.60 (min = 1, max = 40) years. Of the teachers, 212 (32.8%) were primary school teachers, 207 (32.0%) were secondary school teachers, and 228 (35.2%) were high school teachers. In this study, 523 (80.8%) of the teachers were married, and 124 (19.2%) were single (Table 1).

According to the Fitzpatrick skin type, 36 (5.6%) had skin type 1, 163 (25.2%) had skin type 2, 340 (52.6%) had skin type 3, 104 (16.1%) had skin type 4, and 4 (0.6%) had skin type 5. While 544 (84.1%) of the teachers exhibited at least one sun-protection behavior, 103 (15.9%) did not use any sun-protection method (Table 1). The most common of these measures were: avoiding being outside during peak sun hours (between 10 a.m. and 4 p.m.) (n = 493, 76.2%); using sunglasses (n = 420, 64.9%); staying in the shade (n = 416, 64.3%); and using sunscreen products (n = 388, 60.0%) (Table 2).

In our study, 637 (98.5%) participants stated that they had heard of skin cancer, while 10 (1.5%) stated that they had not. Participants most frequently utilized the internet (n = 491, 75.9%) and TV/radio (n = 321, 49.6%) as sources of information on skin cancer, while school was the least likely source (n = 85, 13.1%) (Table 2).

Among the methods of sun protection, women were more likely to prefer not to be outside during hours of intense sun (between 10 a.m. and 4 p.m.), use sunglasses, and use sunscreen products compared to men (*p* < 0.001, *p* < 0.001, *p* < 0.001 respectively). However, no statistical difference existed between the usage rates of the other sun-protection methods (Table 3).

In our survey of teachers, 88.9% of those with Fitzpatrick type 1 skin preferred sunscreen products. A significant correlation was found between skin type and sunscreen use (*p* < 0.001). In addition, women, primary school teachers, those with skin type 1, those with multiple nevi, and those with a high level of SC knowledge used statistically significantly more sunscreen (respectively *p* = 0.001, *p* = 0.003, *p* < 0.001, *p* = 0.037, *p* = 0.002) (Table 3).

The knowledge level of the teachers about SC was 13.54 ± 4.48 (min = 0, max = 23). The knowledge level was 14.03 ± 4.30 for women and 12.66 ± 4.68 for men. The knowledge level of women was found to be significantly higher than that of men (*p* < 0.001). The knowledge level did not differ significantly by age group, marital status, teaching level, or years of service (*p* > 0.05). SC knowledge level was found to be significantly higher in those with a family history of SC and a birthmark on their body (*p* < 0.001, *p* = 0.042, respectively). Those with a higher level of knowledge were more likely to take sun-protection precautions (*p* = 0.032) (Table 1). We found a significant relationship between using sunscreen products, being careful not to be outside during hours of intense sun (between 10 a.m. and 4 p.m.), staying in the shade, using sunglasses, and the level of knowledge. (*p* = 0.002, *p* = 0.028, *p* = 0.024, *p* = 0.042, respectively) (Table 3).

In a survey of SC knowledge, 596 (92.1%) respondents indicated that early diagnosis of SC improved treatment success. Five hundred and fifty-four (85.6%) of them felt that examining one’s skin is vital for spotting symptoms of skin cancer, while 489 (75.6%) said that precautions against SC should begin in childhood. Three hundred and fifty (54.1%) stated that skin cancer is not communicable, 278 (43.0%) reported that skin cancer is not limited to sun-exposed areas, and 201 (31.1%) reported that skin cancer may be easily identified (Table 4).

In our study, 357 (55.2%) of the teachers stated that non-healing wounds, 134 (20.7%) white spots, and 436 (67.4%) nevi that were growing, itching, and bleeding on the skin might be a sign of skin cancer. In addition, 555 (85.8%) of the participants stated that UV rays (sun exposure), 509 (78.7%) stated that exposure to radiation, and 549 (84.9%) stated that exposure to chemical agents could cause skin cancer (Table 4).

Among the SC risk factors, the participants knew the most about smoking (n = 457, 70.6%). On the other hand, the least known factors were having many nevi and freckles, (n = 323, 49.9%) and a light skin type that did not tan at all (Fitzpatrick type 1) (n = 325, 50.2%) (Table 4).

## 4. Discussion

The U.S. Community Preventive Services Task Force has recommended schools as the most appropriate places to improve knowledge and behaviors related to the sun. Therefore, training on this subject has been implemented in primary and secondary schools [29,30].

More recently, studies in Australia have emphasized the importance of reducing exposure to UVR in childhood through teachers. It was emphasized that teachers’ knowledge levels about UVR were unknown and that research should be conducted on this subject [11,24]. After training was given to primary school teacher candidates on SC and UVR, it was reported that their level of knowledge had increased and they felt knowledgeable, talented, and confident enough to train their future students on this subject [24,25,31]. In a six-month study evaluating education in South Carolina using the train-and-equip method for school teachers and school nurses, it was discovered that they transferred what they learned from this training to classroom activities at a high rate (88–100%) [31]. In a study conducted in France with 282 children aged 8–11, it was observed that the knowledge level of the group that received training on skin and sun protection increased significantly after six months; however, there was no change in their sun-protection habits [9]. It has been demonstrated that students whose teachers exhibit sun-protection behaviors take sun-protection precautions. Tanning is popular among students in Turkey, so they may not want to use sunscreen as it will reduce their chances of tanning [20]. For this reason, we think that teachers can impact not only the level of knowledge but also the habits of students with their correct behavior regarding SC prevention.

We concluded that the level of SC knowledge was moderate among teachers. While the knowledge levels of teachers were found to be low in a published study in 1998, it was reported that their knowledge level was moderate in published studies in 2015 and 2022 [21,22,23,26]. These results show that teachers’ awareness of SC has increased recently.

Most of the participants in these studies were women, and we observe that women’s interest in medical aesthetic procedures has increased considerably recently; thus, we deduce that they have recently learned more about skin health. As we expected, female teachers’ knowledge levels on SC were significantly higher than males’ in our study, and the rate of using many sun-protection methods was higher than that of males. Similarly, studies conducted with university students [32,33] and nurses [34,35] revealed that women’s knowledge of SC was significantly higher than men’s. In a study conducted with high school teachers and students, the rate of sunscreen use was found to be higher in female teachers, but there was no difference between genders in students [26]. This study’s results with kindergarten teachers reported no difference in the knowledge level of male and female teachers [23].

The level of knowledge was found to be high in teachers who have SC in their family or have a birthmark. We propose that the level of knowledge is high in these two groups, as these situations lead to the need to obtain information; however, the majority of birthmarks are not associated with SC. There is a very low risk of developing SC from epidermal nevi and an even lower risk from congenital nevi [36]. In studies conducted with nurses and nursing students, there was no statistically significant difference between the presence of a birthmark and the level of SC knowledge [34,35].

There was a positive correlation between the knowledge level of SC, taking precautions to protect from the sun, and the use of sunscreen products. In addition, there was a significant correlation between having skin type 1 and multiple nevi, known as risk factors for SC, and the use of sunscreen. According to the literature, the teachers’ high level of knowledge and/or being in the risk group for SC influenced their sun-protection behaviors positively [26].

The factors underlying the development of SC as the cause and risk factors have been researched in various studies. In these studies, exposure to the sun (31.5%–95.71%) [18,19,21], exposure to chemical agents (49.8–90.6%) [18,37], radiation exposure (44.7–90.6%) [18,37], family history of SC (37.4%–91.43%) [18,19,21,22,30,37], skin type that never tanned (17.3%) [17], light eye color (11.6–71.6%) [17,19,37], and smoking (26.5%–38.4%) [17,18], were found to be SC’s causes or risk factors. In our study, 85.8% of teachers stated that UV rays (sun exposure), 84.9% stated that chemical agents, and 78.7% stated that radiation exposure could cause SC. In our study, the most well-known risk factors were smoking (70.6%) and having SC in the family (65.1%); the least known were having multiple nevi or freckles (49.9%) and light skin and eyes (50.2%). Studies have shown that knowledge of SC causes and risk factors varies widely. This result may be due to the sociodemographic characteristics of the participants (age, gender, education level, occupation, and economic income level), the development level of the education and health systems, and the geographical characteristics of the countries. The most crucial element in the fight against SC is the development of protective behaviors toward risk factors.

The moderate lack of knowledge of teachers about SC was identified in our study. In our study, the participants stated that, respectively, growing, itching, and bleeding nevi, non-healing wounds, and, whitish spots on the skin could be signs of SC (67.4%, 55.2%, and 20.7%, respectively). In Andsoy et al.’s study on nurses; growing, itching, and bleeding nevi (73.5%); non-healing wounds (72.3%); and whitish spots (49.5%) were expressed as SC symptoms, similar to the results of our study [34]. Examining one’s skin is crucial for identifying SC symptoms, according to 85.6% of the teachers. In the study conducted by Celik et al. on nursing students, 91.2% of the participants stated that it is important to examine one’s skin [35]. The knowledge of teachers about SC symptoms and the importance of examining one’s skin is lower than that of nurses and nursing students when our study and these studies’ results are compared. This is likely because these studies were conducted on health workers.

It was seen that a significant percentage of the teachers did not know that SC is not contagious and can occur in places that are not exposed to the sun. As a result of this situation, teachers will pay less attention to skin lesions in areas that are not exposed to the sun. In addition, one out of every four teachers do not know that primary prevention from SC begins in childhood, which will cause them not to make recommendations to protect students and their relatives in this age group (children, nephews, etc.) from SC. Increasing teachers’ knowledge on this subject and transferring this information to their students will enable SC to be recognized in the early stages and treated curatively. In other studies conducted with teachers, their knowledge levels were not reported in terms of SC symptoms [26].

The teachers who participated in our study used the internet as the first source of information and TV/radio as the second. In similar studies in which teachers participated, the internet was preferred as the second and third source of information [26]. However, in our country, this information is provided through organized seminars, displayed posters, and distributed brochures in hospitals [13]. Non-experts providing information on the internet may lead to inappropriate behavior. For this reason, more active information and awareness-raising activities should be carried out in schools, as well as through the internet, social media, and TV and radio by experts. Additionally, health policies should be developed in this direction.

Our study has limitations. First, a non-validated instrument was used in our study. However, the reliability of knowledge questions on SC was assessed using the internal consistency coefficient. The Cronbach alpha value was quite reliable. A second limitation was the cross-sectional design of this study in Kahramanmaras, one of the big cities of Turkey. Therefore, the results cannot be generalized to all teachers in Turkey. Additionally, the cross-sectional design of this study prevented the observation of protective behaviors among teachers. A third limitation of our study may be that teachers’ were not questioned about their working areas. For instance, biology teachers may have more information about UV radiation. Prospective multicenter studies across Turkey are needed to produce more generalizable results.

In conclusion, our study showed teachers’ lack of knowledge about SC and incomplete and wrong orientations in their behaviors toward sun protection. However, the knowledge level of female teachers, teachers with SC in their families, and those with birthmarks was high. We think that the fact that women attach more importance to their skin’s health and that people are worried about SC due to their skin characteristics leads them to learn about SC.

There was a significant relationship between the level of knowledge of SC, having skin type 1, and having plenty of nevi known as a risk factor for SC and the use of sunscreen. The teachers’ high level of knowledge and/or being in the SC risk group influenced their sun-protection behaviors positively.

These results show that increasing the level of knowledge about sun protection and SC will increase sun-protective behaviors. Projects should be carried out by health policymakers to improve the knowledge and behaviors of teachers, and therefore students. Making these projects available on the internet and on TV/radio will allow them to reach a larger audience. Thus, creating such projects will make a great contribution to both public health and health economics.

## Figures and Tables

**Table 1 children-10-00291-t001:** Sociodemographic characteristics of the teachers and their knowledge levels regarding skin properties and skin cancer.

Parameter	Variable	n (%)	Knowledge Level Mean ± SD	*p*
Age (years)	22–30	132 (20.4)	13.17 ± 4.46	0.422
	31–45	382 (59.0)	13.73 ± 4.51	
	45–65	133 (20.6)	13.38 ± 4.42	
Gender	Female	417 (64.5)	14.03 ± 4.30	<0.001 ***
	Male	230 (35.5	12.66 ± 4.68	
Marital status	Married	523 (80.8)	13.54 ± 4.47	0.971
	Single	124 (19.2)	13.53 ± 4.55	
Having children	Yes	492 (76.0)	13.59 ± 4.43	0.601
	No	155 (24.0)	13.38 ± 4.66	
Teaching grade	Primary school	212 (32.8)	13.37 ± 4.56	0.801
	Secondary school	207 (32.0)	13.61 ± 4.46	
	High school	228 (35.2)	13.63 ± 4.43	
Working years	1–15	372 (57.5)	13.65 ± 4.45	0.489
	15–40	275 (42.5)	13.40 ± 4.53	
Plenty nevus	Yes	548 (84.7)	13.53 ± 4.51	0.903
	No	99 (15.3)	13.59 ± 4.36	
Birthmarks	Yes	137 (21.2)	14.78 ± 3.92	<0.001 ***
	No	510 (78.8)	13.21 ± 457	
Family history of skin cancer	Yes	27 (4.2)	15.25 ± 4.33	0.042 *
	No	620 (95.8)	13.47 ± 4.47	
Having had a sunburn before	Yes	315 (48.7)	13.50 ± 4.22	0.835
	No	332 (51.3)	13.58 ± 4.72	
Skin type	Type 1	36 (5.6)	15.69 ± 4.76	0.004 **
	Type 2	163 (25.2)	13.99 ± 4.42	
	Type 3	340 (52.6)	13.34 ± 4.38	
	Type 4	104 (16.1)	12.88 ± 4.59	
	Type 5	4 (0.6)	10.25 ± 7.50	
	Type 6	-	-	
Behaviors of sun protection	Yes	544 (84.1)	13.70 ± 4.44	0.032 *
	No	103 (15.9)	12.67 ± 4.60	

n = frequency, % = column percentage, SD = Standard deviation, * *p* < 0.05; ** *p* < 0.01; *** *p* < 0.001. Type 1: Pale white skin, blue/green eyes, blond/red hair. Always burns do not tan. Type 2: Fair skin, burns easily, tans poorly. Type 3: Darker white skin tans after the initial burn. Type 4: Light brown skin burns minimally, tans easily. Type 5: Brown skin rarely burns, tans darkly easily. Type 6: Dark brown or black skin. Never burns always tans darkly.

**Table 2 children-10-00291-t002:** The information sources of participants about skin cancer and their behaviors of sun protection.

Information Sources	n (%)
Internet	491 (75.9)
Tv-radio	321 (49.6)
Relatives-friends	288 (44.5)
Books-journals	207 (32.0)
Health workers	198 (30.6)
School	85 (13.1)
**Precautions for sun protection**	
I try not to be outside when the sun is intense (between 10 am–4 pm)	493 (76.2)
I stay in the shadows	416 (64.3)
I wear sun-protective clothing	152 (23.5)
I use sunscreen products	388 (60.0)
I wear sunglasses	420 (64.9)
I wear hats	165 (25.5)
I use an umbrella	37 (5.7)

n = frequency, % = column percentage.

**Table 3 children-10-00291-t003:** Status of sociodemographic characteristics according to sun-protection methods.

Sociodemographic Characteristics		Avoid the Sun between 10 a.m.–4 p.m.	Staying in the Shade	Wear Sun-Protective Clothing	Sunscreen Products	Sunglasses	Hat	Umbrella
**Gender**	Male	155 (67.4)	153 (66.5)	51 (22.2)	64 (27.8)	95 (41.3)	69 (30.0)	8 (3.5)
	Female	338 (81.1)	263 (63.1)	101 (24.2)	324 (77.7)	325 (77.9)	96 (23.0)	29 (7.0)
	*p*	<0.001 ***	0.380	0.557	<0.001 ***	<0.001 ***	0.051	0.068
**Marital status**	Married	405 (77.4)	336 (64.2)	130 (24.9)	310 (59.3)	337 (64.4)	139 (26.6)	33 (6.3)
	Single	88 (71.0)	80 (64.5)	22 (17.7)	78 (62.9)	83 (66.9)	26 (21.0)	4 (6.2)
	*p*	0.128	0.955	0.093	0.458	0.601	0.198	0.184
**Teaching grade**	Primary school	169 (79.7)	130 (61.3)	48 (22.6)	147 (69.3)	149 (70.3)	47 (22.2)	10 (4.7)
	Secondary school	159 (76.8)	136 (65.7)	49 (23.7)	118 (57.0)	138 (66.7)	53 (25.6)	14 (6.8)
	High school	165 (72.4)	150 (65.8)	55 (24.1)	123 (53.9)	133 (58.3)	65 (28.5)	13 (5.7)
	*p*	0.189	0.544	0.933	0.003 **	0.026 *	0.313	0.666
**Skin type**	Type 1	83 (70.9)	24 (66.7)	14 (38.9)	32 (88.9)	31 (86.1)	15 (41.7)	4 (11.1)
	Type 2	261 (80.8)	105 (64.4)	37 (22.7)	103 (63.2)	102 (62.6)	49 (30.1)	13 (8.0)
	Type 3	116 (71.6)	219 (64.4)	78 (22.9)	209 (61.5)	232 (68.2)	73 (21.5)	20 (5.9)
	Type 4	33 (73.3)	68 (63.0)	23 (21.3)	44 (40.7)	55 (50.9)	28 (25.9)	0 (0.0)
	*p*	0.346	0.982	0.161	<0.001 ***	<0.001 ***	0.022 *	0.018 *
**Plenty nevus**	Yes	413 (75.4)	346 (63.1)	125 (22.8)	338 (61.7)	356 (65.0)	139 (25.4)	31 (5.7)
	No	80 (80.8)	70 (70.7)	27 (27.3)	50 (50.5)	64 (64.6)	26 (26.3)	6 (6.1)
	*p*	0.242	0.148	0.335	0.037 *	0.951	0.850	0.874
**Family history of skin cancer**	Yes	21 (77.8)	18 (66.7)	8 (29.6)	20 (74.1)	21 (77.8)	9 (33.3)	0 (0.0)
	No	472 (76.1)	398 (64.2)	144 (23.2)	368 (59.4)	399 (64.4)	156 (25.2)	37 (6.0)
	*p*	0.844	0.793	0.442	0.126	0.153	0.340	0.393
**Birthmarks**	Yes	101 (73.7)	92 (67.2)	28 (20.4)	85 (62.0)	94 (68.6)	33 (24.1)	11 (8.0)
	No	392 (76.9)	324 (63.5)	124 (24.3)	303 (59.4)	326 (63.9)	132 (25.9)	26 (5.1)
	*p*	0.444	0.432	0.342	0.577	0.307	0.669	0.190
**Level of knowledge**	Mean ± SD	14.01 ± 4.36	13.86 ± 4.31	14.08 ± 4.31	14.06 ± 4.21	13.88 ± 4.24	13.75 ± 5.06	14.62 ± 4.61
	*p*	0.028 *	0.024 *	0.063	0.002 **	0.042 *	0.296	0.181

n = frequency, % = column percentage, SD = Standart deviation, * *p* < 0.05; ** *p* < 0.01; *** *p* < 0.001.

**Table 4 children-10-00291-t004:** The responses of teachers to knowledge questions.

Statements	I Agree n (%)	I Do Not Agree n (%)	I Have No Idea n (%)
Skin cancer occurs only in sun-exposed areas. (F)	36 (5.6)	278 (43.0)	333 (51.5)
Skin cancer is contagious. (F)	25 (3.9)	350 (54.1)	272 (42.0)
The protection against skin cancer starts in childhood. (T)	489 (75.6)	20 (3.1)	138 (21.3)
Diagnosing skin cancer is easy. (T)	201 (31.1)	64 (9.9)	382 (59.0)
The self-examination of individuals is vital to notice signs of skin cancer. (T)	554 (85.6)	10 (1.5)	83 (12.8)
Skin cancer is less lethal than other types of cancer. (F)	186 (28.7)	133 (20.6)	328 (50.7)
Early detection of skin cancer improves treatment success. (T)	596 (92.1)	6 (0.9)	45 (7.0)
The treatment of skin cancer can only be performed by surgery. (F)	14 (2.2)	211 (32.6)	422 (65.2)
**Leads to skin cancer:**			
Ultraviolet rays (Sun exposure) (T)	555 (85.8)	4 (0.6)	88 (13.6)
Radiation exposure (T)	509 (78.7)	11 (1.7)	127 (19.6)
Exposure to chemicals (T)	549 (84.9)	1 (0.2)	97 (15.0)
**Risk factors for skin cancer**			
Being fair (T) Light-colored eyes (T) Having a never tanning skin type (T)	325 (50.2)	54 (8.3)	268 (41.4)
Dark hair color (F)	13 (2.0)	206 (31.8)	428 (66.2)
Having plenty of nevus and freckles (T)	323 (49.9)	58 (9.0)	266 (41.1)
Having had sunburn (T)	375 (58.0)	42 (6.5)	230 (35.5)
Getting tanned in solariums (T)	373 (57.7)	25 (3.9)	249 (38.5)
Having family members with a history of skin cancer (T)	421 (65.1)	50 (7.7)	176 (27.2)
Having a weak immune system (T)	382 (59.0)	33 (5.1)	232 (35.9)
Smoking (T)	457 (70.6)	22 (3.4)	168 (26.0)
**Signs of the skin cancer**			
Non-healing wounds (T)	357 (55.2)	39 (6.0)	251 (38.8)
White spots (T)	134 (20.7)	60 (9.3)	453 (70.0)
Nevi growing, itching, and bleeding (T)	436 (67.4)	16 (2.5)	195 (30.1)

n = frequency, % = column percentage, T = True, F = False.

## Data Availability

The data supporting this study’s findings are available on request from the corresponding author.

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
