# Peer review of "Knowledge and Protective Behaviors of Teachers on Skin Cancer: A Cross-Sectional Survey Study from Turkey"

_children, 2023, doi:10.3390/children10020291_

Round 1

Reviewer 1 Report

The authors aim at exploring the knowledge, sun protection behaviors and attitudes of teachers towards sun protection and skin cancer. This topic is of relevance, since the educational staff plays an important role in providing sun protection measures and in developing of attitudes toward sun protection in school children. However, it remains unclear what this study adds to the existing body of research.

There are some several flaws in the manuscript that need to be addressed.

Major comments:

-          I strongly recommend proofreading the manuscript by a native speaker

-          Introduction: The introduction needs to be thoroughly revised. At present, the current state of research is missing here. What is already known about knowledge among teachers? What are the research gaps?  Why is this study necessary and useful? Some parts from the discussion can be incorporated in the introduction, e.g., paragraphs 2 and 3

-          Discussion: In the discussion, please work out more clearly what your study adds to the previous state of research. Please also provide a discussion of the limitations of the study (e.g. the use of non-validated instrument, cross-sectional design of your study, response rate, potential biases etc.) and what consequences these limitations have for the interpretations of the results

Minor comments:

Abstract

-          Please provide some more details on sampling technique and the statistical methods used to analyze the data in the abstract

-          Line 29: please specify “sun protection behaviors”

-          Line 35: please be more specific regarding what you mean by “birthmark”. Any specific kind/ particularly high number? There are probably hardly any people without some kind of birthmark in the world

-          The terms “SC knowledge score” and “knowledge level” are used interchangeable in the abstract and throughout the manuscript. Please use consistent terminology throughout the manuscript (e.g. SC knowledge score values?)

-          Conclusion: The first sentence does not follow from the results you present in the abstract. On the basis of which results do you one come to this conclusion (Mean value of knowledge score of 13.5 seems relatively high to me)?. Nothing was reported about the attitudes in the results, but you conclude that attitudes were incomplete/ incorrect

Introduction:

-          The introduction should provide more information on the specific research gap that the study aims to fill. For example, it could mention if previous studies have investigated teachers' knowledge, attitudes, and behaviors related to skin cancer, and if so, what the findings were, and how the current study builds upon or differs from previous research.

-          The introduction should also refer to the specific research question of this study, such as what specific aspects of teachers' knowledge, attitudes, and behaviors the study aims to investigate, and what hypotheses or predictions the study is testing.

-          Line 48: Please specify here the abbreviation “SC”

-          Line 58-59: Something has gone wrong here. This sentence is not clear. Please provide some references for the statement “Changes in exposure to risk factors …”

-          Line 61: What does “on the other hand” refer to?

-          Lines 62-65: This sentence is not clear, please revise

-          Lines 66-68: Please provide references for the sentence “Because children’s skin …”

-          Lines 69-71: Please add references for the statement “Teachers’ role model …”

-          Lines 71-72: What exactly is meant by “behaviors” regarding skin cancer?

Materials and Methods

-          Please provide some more details about the reliability and validity of the questionnaire and the methods that were used to ensure the quality of the data.

Please report the rationale behind using google forms as a data collection method and the response rate.

-          Line 75: What does “(10)” refer to?

-          Line 81: What is meant by “completely”? Only questionnaires containing no missing information were included in the study? Please be more specific here

-          Lines 81-83: Gender distribution and distribution by school type are also shown in the results (and are part of the results), so please delete this sentence and the reference to Table 1 here.

-          Line 86: What sociodemographic aspects were surveyed? Specify why they matter for the topic

-          Line 88: “Presence of nevus and birthmark” -> Please specify, what was meant by these terms or how these data were gathered. Did you ask for specific types of birthmarks/ nevi? See my comment regarding birthmark in the abstract. Please use the revised terminology consistently throughout the manuscript

-          Line 90: I suggest using “Fitzpatrick classification” or “Fitzpatrick skin type” instead of “Fitzpatrick scale” and “Fitzpatrick skin scale”

-          Lines 99-104: please move this part on calculation of the score to the subsection ‘Statistical analysis’

-          Lines 112-113: This statement cannot be correct. Did you mean: “P value <0.05 was defined to be significant”?

Results

-          Line 117: I suggest using “average duration” instead of “average period” (here and hereinafter)

-          Line 125: please use “during peak hours” instead of “peak sun periods” (here and hereinafter)

-          Line 126: please use “sunscreen” or “sunscreen product” instead of “sunscreen cream” (here and hereinafter)

-          Line 133 and hereinafter: Please be careful with using the term “determined” in your manuscript. I suggest “showed” or “reported” instead of it

-          Page 6, line 10: “differ” instead of “change”

-          Lines 19-25: This sentence is too long and somewhat misleading. Please consider to separate it into sentences and clarify, whether the presented numbers refer to the agreement with correct of with incorrect statements

Discussion:

-          The description of how the findings of this study might be used to guide future treatments or policies to enhance teachers' sun safety might be more precise

-          The Discussion could be more specific in terms of how the results of this study could be used to inform future interventions or policies to improve sun safety among teachers, by providing specific examples of interventions that could be implemented and their expected impact

-          Lines 39-41: The sentence is too vague, what do you mean by “related to the sun”?

-          Lines 50-51: please provide references for the statement “many studies have been carried out… “

-          Lines 51-55: please provide references for this sentence

Table 1

-          Please move the table into the Results section

-          In the title, authors write about attitudes. Please provide some more information on how attitudes were assessed in this study

-          Please provide some more details on how the questions/ the questionnaires were validated?

-          “Plenti nevus” -> Please specify in Methods, how it was defined

-          “Having had a sunburn before” -> Please specify in Methods, how did you gather this information. The high number of those, who did not have any sunburn before (51.3%) seems implausible.

-          Skin type -> The definition of skin types is generally known, so that no detailed description is necessary here

Table 2

-          “I wear sunscreen clothes” -> did you mean “sun-protective” clothes?

Table 3

-          “Drink plenty of fluids” does not primarily refer to sun protection, which is the main topic of this manuscript. I suggest removing this column from the table

Table 4

-          How did you made the differentiation between “leads to skin cancer” and Risk factors for skin cancer”? In my opinion, UVR and Radiation exposure are risk factors for SC as well

Author Response

(1) The authors aim at exploring the knowledge, sun protection behaviors and attitudes of teachers towards sun protection and skin cancer. This topic is of relevance, since the educational staff plays an important role in providing sun protection measures and in developing of attitudes toward sun protection in school children.

-Thank you for your comments on our article.

Major comments:

(2) I strongly recommend proofreading the manuscript by a native speaker.

-The manuscript has been edited for English language, grammar, punctuation, spelling, and overall style by one or more Native English editors.

(3) Introduction: The introduction needs to be thoroughly revised. At present, the current state of research is missing here. What is already known about knowledge among teachers? What are the research gaps? Why is this study necessary and useful? Some parts from the discussion can be incorporated in the introduction, e.g., paragraphs 2 and 3.

The literature was reviewed according to your suggestions and questions, and the introduction was rearranged. The paragraphs you suggested in the discussion section have been added to the introduction.

(4) Discussion: In the discussion, please work out more clearly what your study adds to the previous state of research. Please also provide a discussion of the limitations of the study (e.g. the use of non-validated instrument, cross-sectional design of your study, response rate, potential biases etc.) and what consequences these limitations have for the interpretations of the results.

- In the discussion section, similar and different results of our article with similar studies in the literature were evaluated.

"Teachers have incomplete and wrong information and behaviors about SK. As the knowledge about SC increases, correct behaviors are observed to increase. The internet was the most preferred source of information. Information and recommendations should be made by experts via the Internet."

-Limitations of the study were added to the discussion section.

“Our study has limitations. First, a non-validated instrument was used in our study. However, the reliability of knowledge questions on SC was assessed using the Cronbach α internal consistency coefficient. The Cronbach alpha value was quite reliable. A second limitation was the cross-sectional design of this study, in KahramanmaraÅŸ, one of the big cities of Turkey. Therefore, the results cannot be generalized to all teachers in Turkey. The cross-sectional design of this study, which prevented the observation of protective behaviors among teachers. A third limitation of our study may be that teachers' working area was not questioned. Biology teachers may have more information about UV radiation. Prospective multicenter studies across Turkey are needed to produce more generalizable results.”

Minor comments:

(5) Abstract

- Please provide some more details on sampling technique and the statistical methods used to analyze the data in the abstract.

- Line 29: please specify “sun protection behaviors.”

- Line 35: please be more specific regarding what you mean by “birthmark”. Any specific kind/ particularly high number? There are probably hardly any people without some kind of birthmark in the world.

- The terms “SC knowledge score” and “knowledge level” are used interchangeable in the abstract and throughout the manuscript. Please use consistent terminology throughout the manuscript (e.g. SC knowledge score values?).

- Conclusion: The first sentence does not follow from the results you present in the abstract. On the basis of which results do you one come to this conclusion (Mean value of knowledge score of 13.5 seems relatively high to me)?. Nothing was reported about the attitudes in the results, but you conclude that attitudes were incomplete/ incorrect.

- Details about the sample and statistics have been added to the Abstract section. Detailed statistical analysis could not be given due to the word limit in the abstract.

“The number of teachers employed at Kahramanmaras was 18603. Accordingly, the sample was found to be 641, with a 5% margin of error and 99% confidence interval. Schools were selected by simple random method.” and “P value <0.05 was defined to be significant.” sentences added.

- Line 29: Added explanation about “sun protection behaviors”.

“Avoid the sun between 10 am-4 pm, stand in the shade, wear sunscreen clothes, sunglasses, hat, use sunscreen products, and an umbrella are among the methods of protection from the sun in literature.”

- Line 35: -Any color change associated with or not associated with skin cancer seen on the skin at birth is referred to as a birthmark by the public. The first formations that come to mind as birthmarks are café au lait spots, hemangiomas, epidermal nevi, congenital nevi, and mongolian spots. The majority of these are not associated with skin cancer. However, there is a risk of developing skin cancer from epidermal nevi (very rarely) and from congenital nevi. At the same time, since this question was asked in similar studies in the literature, a question about birthmarks was added to our questionnaire. Therefore, an explanation about birthmarks has been added to the discussion part of the article as follows:

"Whereas the majority of birthmarks are not associated with skin cancer. There is a risk of developing skin cancer very rarely from epidermal nevi and rarely from congenital nevi."

(Ryan E, Warren L. Birthmarks--identification and management. Aust Fam Physician. 2012;41(5):274-277.)

- "Knowledge level" terminology was used throughout the manuscript.

- Conclusion: Thank you for your comments on the expression "attitude". We revised the conclusion section. We realized that we were using the term "attitude" incorrectly. We have removed this statement from the title and article. The teacher's knowledge level was interpreted as moderate by reviewing the literature.

“The knowledge level of teachers about skin cancer and sun protective behaviors was moderate. With the increase in knowledge about SC, correct behaviors increase. Information and recommendations made on the Internet should be made by experts. Implementing projects aimed at improving teachers' knowledge and behaviors, through them, students about SC by health policymakers will significantly contribute to both public health and the health economics.”

(6) Introduction:

- The introduction should provide more information on the specific research gap that the study aims to fill. For example, it could mention if previous studies have investigated teachers' knowledge, attitudes, and behaviors related to skin cancer, and if so, what the findings were, and how the current study builds upon or differs from previous research.

- The introduction should also refer to the specific research question of this study, such as what specific aspects of teachers' knowledge, attitudes, and behaviors the study aims to investigate, and what hypotheses or predictions the study is testing.

- Line 48: Please specify here the abbreviation “SC”

- Line 58-59: Something has gone wrong here. This sentence is not clear. Please provide some references for the statement “Changes in exposure to risk factors …”

- Line 61: What does “on the other hand” refer to?

- Lines 62-65: This sentence is not clear, please revise

- Lines 66-68: Please provide references for the sentence “Because children’s skin …”

- Lines 69-71: Please add references for the statement “Teachers’ role model …”

- Lines 71-72: What exactly is meant by “behaviors” regarding skin cancer?

- The introduction section has been reorganized.

“Turkey is one of the countries where public health policies for sun safety, which is the most important preventable risk factor for skin cancer, have not yet been established. In the literature, studies evaluating the knowledge and behaviors of teachers and preservice teachers about skin cancer and sun safety were mostly conducted in countries where these public health policies were developed.

We came across two studies conducted for teachers on this subject, in our country. Studies to be made on this subject will guide policies to improve solar safety for countries like our country, where policies have not yet been developed in this regard and, where year-round UV index averages are high. This study aimed to evaluate evaluate the knowledge level of teachers about SC and their sun-protective behaviors.”

- The introduction section has been reorganized. The aim of the study was detailed.

“This study aimed to evaluate evaluate the knowledge level of teachers about SC and their sun-protective behaviors.”

- Line 48: The abbreviation “SC” was added as skin cancer.

- Line 58-59: This sentence has been revised and added references.

“Exposure to risk factors, use of screening tests, and treatment modalities affect cancer incidence and cancer-related deaths.”

- Line 61: The phrase “on the other hand” has been deleted.

- Lines 62-65: This sentence has been revised.

“The most effective way to reduce SC risk is to protect against the sun, a significant risk factor. Avoid the sun between 10 am-4 pm, stand in the shade, wear sunscreen clothes, sunglasses, hat, use sunscreen products, and an umbrella are some of the sun protection methods.”

- Lines 66-68: Added references.

- Lines 69-71: Added references.

- Lines 71-72: The participants were asked “Avoid the sun between 10 am-4 pm, stand in the shade, wear sunscreen clothes, sunglasses, hat, use sunscreen products, and an umbrella are some of the sun protection methods”.

(6) Materials and Methods

- Please provide some more details about the reliability and validity of the questionnaire and the methods that were used to ensure the quality of the data.

Please report the rationale behind using google forms as a data collection method and the response rate.

- Line 75: What does “(10)” refer to?

- Line 81: What is meant by “completely”? Only questionnaires containing no missing information were included in the study? Please be more specific here

- Lines 81-83: Gender distribution and distribution by school type are also shown in the results (and are part of the results), so please delete this sentence and the reference to Table 1 here.

- Line 86: What sociodemographic aspects were surveyed? Specify why they matter for the topic

- Line 88: “Presence of nevus and birthmark” -> Please specify, what was meant by these terms or how these data were gathered. Did you ask for specific types of birthmarks/ nevi? See my comment regarding birthmark in the abstract. Please use the revised terminology consistently throughout the manuscript.

- Line 90: I suggest using “Fitzpatrick classification” or “Fitzpatrick skin type” instead of “Fitzpatrick scale” and “Fitzpatrick skin scale”

- Lines 99-104: please move this part on calculation of the score to the subsection ‘Statistical analysis’

- Lines 112-113: This statement cannot be correct. Did you mean: “P value <0.05 was defined to be significant”?

- The methodology of the study was organized in more detail.

“The number of teachers employed at Kahramanmaras was 18603. Accordingly, the sample was found to be 641, with a 5% margin of error and 99% confidence interval. 647 teachers participated in the study. Schools were selected by simple random method. Teachers in 30 schools in Kahramanmaras took part in the study. Based on the literature review on skin cancer, the researchers created a 40-question and three-part questionnaire. The questionnaire was applied to 25 people and was rearranged according to the feed-back. Afterwards, the questionnaire was applied to the teachers via google forms mail method. Surveys were implemented with google forms as it would be difficult to reach for geographical reasons. Before the study, teachers were informed via e-mail and telephone applications. A consent form and questionnaire prepared via Google forms were sent to those who wanted to take part in the study. Teachers who gave consent and completed the questionnaire were included in the study. Teachers who gave incomplete answers to the skin cancer knowledge questions were excluded from the study.”

- Line 75: The phrase “(10)” has been deleted.

- Line 81: “Teachers who gave consent and filled outcompleted the questionnaire were included in the study. Teachers who gave incomplete answers to the skin cancer knowledge questions were excluded from the study.”

- Lines 81-83:"A total of 647 teachers, including 212 (32.8%) primary school teachers, 207 (32.0%) secondary school teachers and 228 (35.2%) high school teachers, participated in the research." sentence was deleted.

- Line 86: In terms of sociodemographic characteristics, age, gender, marital status and whether they have children were questioned by the teachers. Marriage and having children were added to the survey, thinking that they would need to learn more about sun protection. Sociodemographic characteristics were detailed in the article.

“Sibling” was written due to mistranslation. Fixed “Sibling” to “having children”.

- Line 88: Any color change associated with or not associated with skin cancer seen on the skin at birth is referred to as a birthmark by the public. The first formations that come to mind as birthmarks are café au lait spots, hemangiomas, epidermal nevi, congenital nevi, and mongolian spots. The majority of these are not associated with skin cancer. However, there is a risk of developing skin cancer from epidermal nevi (very rarely) and from congenital nevi. At the same time, since this question was asked in similar studies in the literature, a question about birthmarks was added to our questionnaire. Therefore, an explanation about birthmarks has been added to the discussion part of the article as follows:

"Whereas the majority of birthmarks are not associated with skin cancer. There is a risk of developing skin cancer very rarely from epidermal nevi and rarely from congenital nevi."

(Ryan E, Warren L. Birthmarks--identification and management. Aust Fam Physician. 2012;41(5):274-277.)

- Line 90 and others: Changed the statement “Fitzpatrick scale” to “Fitzpatrick classification” or “Fitzpatrick skin type”.

- Lines 99-104: The calculation part of the questionnaire was moved to the subsection 'Statistical analysis'.

- Lines 112-113: "Cases with a P value below 0.05 were considered statistically significant results." sentence was deleted. Instead of "P value <0.05 was defined to be significant." sentence was added.

(7) Results

- Line 117: I suggest using “average duration” instead of “average period” (here and hereinafter)

- Line 125: please use “during peak hours” instead of “peak sun periods” (here and hereinafter)

- Line 126: please use “sunscreen” or “sunscreen product” instead of “sunscreen cream” (here and hereinafter)

- Line 133 and hereinafter: Please be careful with using the term “determined” in your manuscript. I suggest “showed” or “reported” instead of it

- Page 6, line 10: “differ” instead of “change”

- Lines 19-25: This sentence is too long and somewhat misleading. Please consider to separate it into sentences and clarify, whether the presented numbers refer to the agreement with correct of with incorrect statements.

- Line 117: Changed the statement “average period” to “average duration”.

- Line 125 and others: Changed the statement “peak sun periods” to “during peak sun hours”.

- Line 126 and others: Changed the statement “sunscreen cream” to “sunscreen product”.

- Line 133 and others: Changed the statement “determined” to “showed” or “reported”.

- Page 6, line 10: Changed the statement “change” to “differ”.

- Lines 19-25: Sentences were rearranged.

“In the poll that examined the degree of knowledge on SC, 596 (92.1%) respondents indi-cated that early diagnosis of DK boosted treatment success. 554 (85.6%) of them felt that examining one's own skin is vital for spotting skin cancer symptoms, while 489 (75.6%) said that precautions against DK should begin in childhood. 350 (54.1%) stated that skin cancer is not communicable, 278 (43.0%) reported that skin cancer is not limited to solar-exposed areas, and 201 (31.1%) reported that skin cancer may be easily identified (Table 4).”

(8) Discussion:

- The description of how the findings of this study might be used to guide future treatments or policies to enhance teachers' sun safety might be more precise

- The Discussion could be more specific in terms of how the results of this study could be used to inform future interventions or policies to improve sun safety among teachers, by providing specific examples of interventions that could be implemented and their expected impact.

- Lines 39-41: The sentence is too vague, what do you mean by “related to the sun”?

- Lines 50-51: please provide references for the statement “many studies have been carried out… “

- Lines 51-55: please provide references for this sentence.

-The discussion was rearranged according to the suggestions.

Projects should be carried out by health policy makers to improve the knowledge and behaviors of teachers and students through teachers. Making these projects via the internet and TV-radio will help reach wider audiences.”

-The discussion was rearranged according to the suggestions.

“More active information and awareness-raising activities should be carried out in schools, as well as through internet-social media and TV-radio by experts. Health policies should be developed in this direction.”

- Changed the statement “related to the sun” to “behaviors related to sun protection”. (Line 39-41)

- Added source for " many studies have been carried out… " statement. (Lines 50-51)

“Perez D, Kite J, Dunlop SM, Cust AE, Goumas C, Cotter T, et al. Exposure to the 'Dark Side of Tanning' skin cancer prevention mass media campaign and its association with tanning attitudes in New South Wales, Australia. Health Educ Res. 2015;30(2):336-46.”

- Added source for this sentence. (Lines 51-55)

(9) Table 1

- Please move the table into the Results section.

- In the title, authors write about attitudes. Please provide some more information on how attitudes were assessed in this study.

- Please provide some more details on how the questions/ the questionnaires were validated?

- “Plenti nevus” -> Please specify in Methods, how it was defined.

- “Having had a sunburn before” -> Please specify in Methods, how did you gather this information. The high number of those, who did not have any sunburn before (51.3%) seems implausible.

- Skin type -> The definition of skin types is generally known, so that no detailed description is necessary here.

-Table 1 was moved to the Results section of the article.

- We realized that we were using the term "attitude" incorrectly. We have removed this statement from the title and article.

- Detailed explanations about the survey have been added to the methodology section.

“Based on the literature review on skin cancer, the researchers created a 40-question and three-part questionnaire. The questionnaire was applied to 25 people and was rearranged according to the feedback. Afterwards, the questionnaire was applied to the teachers via google forms mail method.”

- Added explanation for plenty nevus to the methodology section.

"Plenty nevus" was defined as the presence of more than 50 nevi.”

-The phrase "Sunburn story" has been added to the method. Teachers may only have answered based on the critical sunburn story. Due to self-reported questionnaires, no comment can be made on this result.

-The explanation given for "skin type" in the table has been deleted.

(10) Table 2

- “I wear sunscreen clothes” -> did you mean “sun-protective” clothes?

- The phrase “Sun-protective clothing” has been deleted and fixed as “wear sunscreen clothes”.

(11) Table 3

- “Drink plenty of fluids” does not primarily refer to sun protection, which is the main topic of this manuscript. I suggest removing this column from the table.

-The “Drink plenty of fluids” column has been removed from the table.

(12) Table 4

- How did you made the differentiation between “leads to skin cancer” and Risk factors for skin cancer”? In my opinion, UVR and Radiation exposure are risk factors for SC as well.

-UVR exposure is a preventable cause of skin cancer. UVR can be also considered as a risk factor. However, personal characteristics and habits are included in the risk factor category.

The list has been revised to include referee comments and answers, edited due to additions to the references. Corrections made are marked in red in the manuscript.

Thank you for your time and assistance,

Best Regards.

Reviewer 2 Report

This is an interesting article that would be worthwhile adding to the literature. 

Few minor comments:

- Can authors clarify the significance of why they added birthmark as a parameter in the questionnaire? Birthmark is quite a broad and non-specific term; and most birth marks, such as vascular birthmarks, are not generally linked to skin cancer.

- Please add information on the range of UV Indices of this region in Turkey throughout the year. For example, is UV Index above 3 in winter? This has implications on whether sun protection should be employed year-round or not. Obviously, there is also the issue of public misunderstanding about ambience temperature not being equal to UV index.

- Can authors comment on the cultural perception of tanning in Turkey? Over half of your cohort is skin type 3 and above. In the Western world, tanning is culturally popular; whereas this is the opposite in most Asian countries where "whiteness" is desirable. This has huge implications on sun-seeking behaviours and should be discussed in your article.

- Please comment on what, if any, public health campaign already existed on skin cancer and sun awareness in Turkey. For example, the SunSmart campaign in Australia is omnipresent in schools and other places since the 80s.

- There is no mention about acral melanoma in the text or the questionnaire, which is not traditionally associated with UV. This is important for this cohort of darker skin types.

- English language editing may help to improve overall readability throughout the article.  For example, the term "Health economics" should be used instead of Health economy. Typo in line 138.

Author Response

Thank you for your interest and comments on our article (ID: children-2155860).

We made the revisions according to your suggestions, and we think our article is better thanks to you. Article files with corrections have been uploaded to the journal system.

Below are the responses to the referees' comments.

Reviewer #1:

(1) This is an interesting article that would be worthwhile adding to the literature.

-Thank you for your comments on our article.

(2) Can authors clarify the significance of why they added birthmark as a parameter in the questionnaire? Birthmark is quite a broad and non-specific term; and most birth marks, such as vascular birthmarks, are not generally linked to skin cancer.

-Any color change associated with or not associated with skin cancer seen on the skin at birth is referred to as a birthmark by the public. The first formations that come to mind as birthmarks are café au lait spots, hemangiomas, epidermal nevi, congenital nevi, and mongolian spots. The majority of these are not associated with skin cancer. However, there is a risk of developing skin cancer from epidermal nevi (very rarely) and from congenital nevi. At the same time, since this question was asked in similar studies in the literature, a question about birthmarks was added to our questionnaire. Therefore, an explanation about birthmarks has been added to the discussion part of the article as follows:

"Whereas the majority of birthmarks are not associated with skin cancer. There is a risk of developing skin cancer very rarely from epidermal nevi and rarely from congenital nevi."

(Ryan E, Warren L. Birthmarks--identification and management. Aust Fam Physician. 2012;41(5):274-277.)

(3) Please add information on the range of UV Indices of this region in Turkey throughout the year. For example, is UV Index above 3 in winter? This has implications on whether sun protection should be employed year-round or not. Obviously, there is also the issue of public misunderstanding about ambience temperature not being equal to UV index.

-The average temperature information of our region in the material-method section has been removed, and the average of the Uv index of our region and general information about the Uv index has been added to the introduction and discussionsection.

-The following sentences are added to the introduction:

"The ultraviolet (UV) index measures the intensity of UV radiation, a measurement that has been used for about 20 years. UV index level exposed is associated with the risk of sunburn and skin cancer."

(Schmalwieser AW, Grobner J, Blumthaler M, et al. UV Index monitoring in Europe. Photochem Photobiol Sci. 2017;16(9):1349-1370. doi:10.1039/c7pp00178a)

(Kaundinya T, Kundu RV, Feinglass J. The epidemiology of skin cancer by UV index: a cross-sectional analysis from the 2019 behavioral risk factor surveillance survey [published online ahead of print, 2022 Jan 8]. Arch Dermatol Res. 2022;10.1007/s00403-021-02313-z. doi:10.1007/s00403-021-02313-z)

"When the UV index is three and above, there is a need to take sunscreen precautions."

(Silva AA. The Shadow Rule, the UV Index, and the 5S Steps in the Tropics. Health Phys. 2020;119(3):358-362. doi:10.1097/HP.0000000000001220)

-The following sentences are added to the discussion:

"In our region, the average UV index in 2020 is 7-8 in summer, 5 in spring, and 2 in winter. Although there is no need for strict sun protection measures in winter in our region, the UV index is above three at noon on sunny days in winter. Therefore, those with skin cancer risk factors and photosensitivity should take sunscreen precautions these days and hours."

(Annual average. Available online: https://www.worldweatheronline.com (accessed on 13.01.2023)

(Silva AA. The Shadow Rule, the UV Index, and the 5S Steps in the Tropics. Health Phys. 2020;119(3):358-362. doi:10.1097/HP.0000000000001220)

(4) - Can authors comment on the cultural perception of tanning in Turkey? Over half of your cohort is skin type 3 and above. In the Western world, tanning is culturally popular; whereas this is the opposite in most Asian countries where "whiteness" is desirable. This has huge implications on sunseeking behaviours and should be discussed in your article.

-Tanning is popular, especially among young people in our country. This information has been added and discussed in the Discussion section.

"Tanning is popular among students in Turkey, so they may not want to use sunscreen as it will reduce their chances of tanning."

(Filiz TM, Cinar N, Topsever P, Ucar F. Tanning youth: knowledge, behaviors and attitudes toward sun protection of high school students in Sakarya, Turkey. J Adolesc Health. 2006;38(4):469-471. doi:10.1016/j.jadohealth.2005.01.016)

(5) Please comment on what, if any, public health campaign already existed on skin cancer and sun awareness in Turkey. For example, the SunSmart campaign in Australia is omnipresent in schools and other places since the 80s.

-There is no comprehensive public health policy in Turkey as in Austria. However, small-scale studies are conducted especially by the Turkish Dermatology Association, to provide information on this subject. More detailed information about the work has been added to the discussion part of the article.

"In Turkey, although not as comprehensive as in Australia, to inform is given on this subject. Every year in Turkey, May has been accepted as the month of skin cancer awareness, posters for this are displayed in hospitals this month, and seminars are held, although usually limited to hospital and provincial health directorate employees. In addition, information brochures on sun protection, and self-examination of the nevus and skin cancer, organized by the Turkish Dermatology Association, can be accessed on the association's website."

(Annual average. Available online: https://www.worldweatheronline.com (accessed on 13.01.2023)

(6) There is no mention about acral melanoma in the text or the questionnaire, which is not traditionally associated with UV. This is important for this cohort of darker skin types.

-Since the questionnaire was designed for teachers, subtypes of melanoma and other skin cancers were not mentioned. The term skin cancer covered all of them so that teachers could understand them more easily. In a further study, informing the participants and mentioning this detail may enable us to obtain better results.

(7) English language editing may help to improve overall readability throughout the article. For example, the term "Health economics" should be used instead of Health economy. Typo in line 138.

-The manuscript has been edited for English language, grammar, punctuation, spelling, and overall style by one or more Native English editors.

Round 2

Reviewer 1 Report

Thank you very much for the revision of the manuscript. I still have some points that need to be changed in the manuscript:

My comment regarding “sunscreen clothing” was misunderstood. I would recommend using the term “sun protective clothing” throughout the manuscript and in the abstract

I recommend using the term “staying in the shade” throughout the manuscript instead of “standing”

Lines 29-30: please rephrase the sentence into: “In this cross-sectional study conducted between September 21, 2020, and October 21, 2020, 647 teachers from 30 schools in Kahramanmaras were included with their consent”

Line 35: Since you do not provide any information on statistical analyses in the Methods, please exclude the sentence on p-values

Line 59: please rephrase the sentence: In Turkey, the SC was diagnosed in 1622 people in 2018 ...

Line 126 and in throughout the manuscript: the term “questioned” seems not to be suitable in this context. Consider using: gathered/ asked/ researched/ reported instead

Line 127: please provide details on how you gathered “sun protection methods” (the frequency of using sunscreen, clothing, umbrella etc? During which period of time or on a certain day or usually?) and “sunburn” (Did you ask for having had sunburn ever in life? sunburn within the last 12 months?)

Table 1: Please delete the details on how the skin types were defined in the second column of the table, i.e. “Pale white skin, blue/green eyes, blond/red hair. Always burns do not tan” etc. It is sufficient if you write just Type 1, Type 2 etc here

Pg 9, line 112: The term “portion” is not suitable here, please use “percentage” instead

Author Response

Thank you for your interest and comments on our article (ID: children-2155860).

We made the revisions according to your suggestions, and we think our article is better thanks to you. Article files with corrections have been uploaded to the journal system.

Below are the responses to the referees' comments.

Reviewer #1:

(1) Thank you very much for the revision of the manuscript. I still have some points that need to be changed in the manuscript.

-Thank you for your comments on our article.

(2) My comment regarding “sunscreen clothing” was misunderstood. I would recommend using the term “sun protective clothing” throughout the manuscript and in the abstract

- Changed the statement “sunscreen clothing” to “sun-protective clothing”.

(3) I recommend using the term “staying in the shade” throughout the manuscript instead of “standing”

- Changed the statement “standing in the shade” to “staying in the shade” throughout the manuscript.

(4) Lines 29-30: please rephrase the sentence into: “In this cross-sectional study conducted between September 21, 2020, and October 21, 2020, 647 teachers from 30 schools in Kahramanmaras were included with their consent.”

-Thank you for your comments. This sentence has been revised according to your comment.

“In this cross-sectional study conducted between September 21, 2020, and October 21, 2020, 647 teachers from 30 schools in Kahramanmaras were included with their consent.”

(5) Line 35: Since you do not provide any information on statistical analyses in the Methods, please exclude the sentence on p-values.

-The sentence on p-values was excluded from the abstract.

(6) Line 59: please rephrase the sentence: In Turkey, the SC was diagnosed in 1622 people in 2018 ...

- The sentence was edited.

“In Turkey, the 5-year prevalence of skin cancer is 4809. There were 1622 new cases and 669 deaths related to SC in 2018.”

(7) Line 126 and in throughout the manuscript: the term “questioned” seems not to be suitable in this context. Consider using: gathered/ asked/ researched/ reported instead

- Changed the statement “questioned” to “gathered/ asked/ researched/ reported” throughout the manuscript.

(8) Line 127: please provide details on how you gathered “sun protection methods” (the frequency of using sunscreen, clothing, umbrella etc? During which period of time or on a certain day or usually?) and “sunburn” (Did you ask for having had sunburn ever in life? sunburn within the last 12 months?)

- The sentence was edited as “sun protection methods on sunny days, sunburn ever in life”

(10) Table 1: Please delete the details on how the skin types were defined in the second column of the table, i.e. “Pale white skin, blue/green eyes, blond/red hair. Always burns do not tan” etc. It is sufficient if you write just Type 1, Type 2 etc here

- Deleted the details on how the skin types were defined in the second column of the table.

(11) Pg 9, line 112: The term “portion” is not suitable here, please use “percentage” instead

- Changed the statement “portion” to “percentage”.

The list has been revised to include referee comments and answers, edited due to additions to the references. Corrections made are marked in red in the manuscript.

Thank you for your time and assistance,

Best Regards.